# OpenReview forum: "Century: A Framework and Dataset for Evaluating Historical Contextualisation of Sensitive Images"
_ICLR.cc/2025/Conference — ICLR 2025 Spotlight_

### Official Review · Reviewer_qqDa · 2024-10-30

**Soundness:** 3
**Presentation:** 3
**Contribution:** 3
**Rating:** 8
**Confidence:** 3

**Summary:**

The authors present Century, a dataset of 1,500 sensitive historical images curated from recent history. It is generated using an automated process that combines knowledge graphs and language models, guided by criteria from museum and digital archive practices to ensure a balanced representation of global events and figures. The dataset is validated through both automated and human evaluations, demonstrating its diversity and comprehensiveness. Additionally, the authors introduce an evaluation framework to measure historical contextualization along dimensions of accuracy, thoroughness, and objectivity, applying it to assess the performance of four foundational models, with both automated metrics and human feedback supporting the results.

**Strengths:**

The paper is well-articulated and clear, enhancing readability and accessibility.

Addressing sensitive historical images is a compelling topic with high relevance, and the proposed framework is both innovative and thoughtfully executed.

The methodology for identifying and curating sensitive historical images, integrating knowledge graphs with language models, provides a scalable approach with potential research applications across history and AI.

The Century dataset could serve as a valuable resource for researchers working on similar challenges, including those focused on historical image representation, automated content generation, and bias mitigation.

**Weaknesses:**

I'm a bit concerned about the dataset scale. At 1,500 images, the dataset may be too small to train deep learning models directly, potentially limiting its use in large-scale AI training scenarios. A dataset size of more than 10K images would be a good estimation for training models.

Furthermore, as a new framework, the effectiveness of Century could benefit from comparative analysis with existing datasets or similar historical image frameworks. This would provide a clearer benchmark of its strengths and limitations. If there are not closer frameworks, some related research might also help in comparison, such as the following papers for your reference:

Wu, Mingfang, et al. "Automated metadata annotation: What is and is not possible with machine learning." Data Intelligence 5.1 (2023): 122-138.

Wadhawan, Rohan, et al. "ConTextual: Evaluating Context-Sensitive Text-Rich Visual Reasoning in Large Multimodal Models." arXiv preprint arXiv:2401.13311 (2024).

Finally, the authors candidly discuss certain biases, particularly concerning dataset distribution and generative labeling. These limitations could impact future applications, and additional mitigative strategies would strengthen the framework's applicability.

Minor: It is unclear to me whether a dataset-centric paper with a focus on historical content aligns fully with ICLR’s primary scope, which typically emphasizes innovations in machine learning.

**Questions:**

Have the authors conducted any formal bias testing within the dataset? Is it possible to elaborate on potential approaches the authors have considered for addressing these biases. Understanding how these biases may clarify the power of the dataset, the impact of model outcomes, and outlining potential mitigation strategies, would further enhance the dataset’s robustness for future research.

Have the authors considered ways to expand the dataset or if they envision it being used primarily for evaluation rather than training.

---

> ### Author Response · Authors · 2024-11-15
> **Response to Reviewer qqDa**
>
> We thank the reviewer for their thoughtful comments on our manuscript.
>
> We note the reviewer’s concern on the dataset scale. We designed the dataset for use in evaluations of historical contextualisation capabilities of multi-modal models - not for model training or fine-tuning.  With this dataset, developers will be able to assess  how well models contextualise sensitive historical images. They can use Century to inform post-training and deployment decisions
> While we invite future work on improving system capabilities, the dataset does not include "target" responses of how systems responded to the different classes of queries for the images in this dataset (ie. what is being evaluated in Table 3). We only release the images and related metadata - a point we make clearer in the updated draft (lines 320-323).  We do hope that the dataset enables application and system developers to make their own ethical and normative choices in their deployment context (possibly adapting or building on our work).
>
> In constructing our evaluation task, we developed several criteria along which we test for the “goodness” of the historical contextualisation response. Surprisingly, even though models are almost certainly exposed to images and their contexts in training, they perform poorly even on fairly objective elements of the task such as identification. This can be clearly seen in the accuracy results we provided in Table 3 of our manuscript where we showed that the best model achieved 53.3% accuracy on this task. Because of this and the other empirical results in Table 3, we don't think training on “target” responses would be likely to lead to improvements along even the accuracy dimensions of our criteria, let alone others that we evaluated in this work.
>
> We cite evaluation frameworks related to historical contextualisation from images in the Related Works section, but to our knowledge, Century is the first benchmark focused on evaluating multi-modal generative models on historical contextualization of images. We have added the two suggested citations on automated metadata annotation and evaluating visual reasoning in context-rich scenarios in our discussion of how Century builds on previous efforts to measure visual reasoning and contextualisation (line 473, 475).
>
> We understand the reviewer’s  concerns on the potential bias in the benchmark. The most thorough analyses of potential biases in Century can be found in Table 8 in Appendix K, where we look at the distribution of figures, events, and locations represented in the dataset by UN world subregion. We find that as per the human evaluation labels, every subregion is represented by at least 5.1% of the images in Century, indicating that our dataset contains a relevant amount of images from all areas of the world. Unsurprisingly, we also find that images taken from Wikipedia are skewed in their distribution (with images from Western Europe and North Americas most prominently featured), but we do find evidence that Century contains images from all areas of the world.
>
> We make recommendations on mitigation strategies that could be applied in future work in the Limitations, calling upon researchers to integrate participatory perspectives and build upon the representativeness of the work in a targeted way (e.g. contributing images of a specific culture). We also are excited to see future work build on our methods for creating the dataset, and adapt them to other historical data sources (eg, cultural archives).
>
> Regarding the fit of this paper in ICLR, we are responding to the call for papers, which explicitly seeks work related to "generative models" and "datasets and benchmarks" and "societal considerations including fairness, safety, privacy."  Our work directly speaks to multiple subject areas in the CFP.  Additionally, previously accepted papers in ICLR 2023 and 2024 include papers that center contributions through datasets and benchmarks:
> * MIntRec2.0: A Large-scale Benchmark Dataset for Multimodal Intent Recognition and Out-of-scope Detection in Conversations, ICLR 2024
> * SWE-Bench: Can Language Models Resolve Real-World GitHub Issues?, ICLR 2024 (oral)
> * MEDFAIR: Benchmarking Fairness for Medical Imaging, ICLR 2023 (notable-top-25%)
>
> We thank the reviewer for their careful engagement with the paper and their suggestions to strengthen the manuscript, which we have included in the updated version. If we have addressed the reviewer’s most pressing concerns, we kindly ask the reviewer to consider adjusting their score to reflect this. Otherwise, we are looking forward to proceeding with the discussion and incorporating any further feedback to our manuscript.

---

> > ### Comment · Reviewer_qqDa · 2024-11-23
> >
> > Thanks for the author feedback. Most my concerns have been addressed. I'd like to raise the rating to accept.

---

### Official Review · Reviewer_18jS · 2024-11-02

**Soundness:** 4
**Presentation:** 4
**Contribution:** 4
**Rating:** 8
**Confidence:** 3

**Summary:**

This paper introduces Century, a dataset with 1,500 images of sensitive historical images (including a new method to identify images like those in the dataset). Along with Century, the authors propose an evaluation framework to measure how well models do at “historical contextualization.”

**Strengths:**

S1. The authors tackle a problem that many researchers shy away from or do not even consider, as historical contextualization is a complex task and has no objective ground truth. This paper is a thorough, high-quality effort to 1) help understand our models through this lens, and 2) highlight the importance of historical contextualization abilities in large vision-language models.

S2. The paper is very well-written; the methods and results are presented in a straightforward manner and thoroughly-discussed.

S3. Century is a diverse dataset with a decent balance across regions, content, and image type. The dataset can always be more diverse and balanced along these axes, but it is a respectable collection for evaluation given that its limitations are acknowledged.

**Weaknesses:**

W1. The evaluations are done on closed-source models, which are helpful in illuminating their capabilities given that we don’t know much about their data or architecture. However, it would be incredibly useful to benchmark open-source VLMs alongside them, as the associations with training data, architecture, etc. and historical contextualization abilities can help the community to identify how to score better on this benchmark.

W2. I would love to see a more thorough limitations section. While the points covered are valid and important, there is so much nuance to the dataset, evaluation metrics, etc. The community would benefit from a paper that not only presented a useful dataset and benchmark for historical contextualization, but thoroughly (to a best approximation) enumerated the pitfalls one could fall into when maximizing performance on this benchmark, and described the demographic and geographic distribution of human evaluators.

W3. Some of the figures seem to be missing legends, or at least are not clear enough in what the colors mean (Figures 6 and 11). I assume the x-axis is labeled 1-5, but the colors and lack of x-axis label are a bit confusing.

**Questions:**

Q1. Is it possible to recover the geographic and demographic distribution of the human evaluators? That data seems especially important to consider for historical contextualization.

---

> ### Author Response · Authors · 2024-11-26
> **Response to Reviewer 18jS**
>
> We appreciate the reviewer's thoughtful feedback on our manuscript. We're pleased they recognised the paper's comprehensive approach to addressing a significant evaluation challenge using a diverse dataset.
>
> Regarding the reviewer's suggestion to evaluate open-source models, we agree that this would be valuable for the community. However, most open-source multimodal models (like PaliGEMMA) require substantial fine-tuning to handle complex tasks like historical contextualization. The few systems that are fit for this purpose are not available for use due to their licenses, so we were unable to evaluate these systems directly. To encourage open investigation, we've made our dataset and evaluation methods publicly available, enabling researchers to explore how current and future systems (including new tokenization strategies, inference-time compute methods, and composite systems) impact historical contextualisation capabilities.
>
> As recommended, we've expanded the limitations section (lines 502-506; 513-517; 524-530) to address potential unintended consequences of our methodology. We also discuss short-term mitigations for developers to avoid potential pitfalls when optimizing for Century, alongside longer-term improvements.
>
> We've added clarification to the captions of Figures 6 (page 23) and 11 (page 34) to improve their interpretation, addressing the reviewer's concerns about the figures.
>
> Regarding demographic data for raters, we are required to store rater data with obfuscated identifiers, which means demographic data for individual raters are not easily recoverable. We acknowledge this as an important addition to future work, especially participatory work that seeks to identify the perspectives of a specific group of people on the historical contextualisation task. We provide details on our the recruitment strategy decisions that likely influenced ater pool composition in lines 305-307 and Appendices L and T.
>
> We thank the reviewer for their careful review and insightful comments, which have strengthened our manuscript.

---

> > ### Comment · Reviewer_18jS · 2024-11-26
> > **thank you**
> >
> > Dear authors, thank you for your thorough response. I am happy with the clarifications and the submission is still in line with my original score of accept.

---

### Official Review · Reviewer_zRQb · 2024-11-03

**Soundness:** 3
**Presentation:** 2
**Contribution:** 2
**Rating:** 6
**Confidence:** 4

**Summary:**

The paper presents a new dataset for evaluating multimodal models’ capability to describe historical and sensitive images in terms of several criteria, including factual errors and due weight. The images in the dataset are carefully chosen so that they are sensitive, controversial, and/or commemorative. The evaluation protocol includes automated evaluation and human evaluation. The paper gives some recommendations for evaluating models with the dataset.

**Strengths:**

I think this evaluation is important conceptually and in the application level. One expectation to a foundation model may be to generate unbiased (or not one-sided) descriptions of sensitive events, and the proposed dataset can serve as a benchmark in this regard.

Also, the paper recommends that human evaluation is still critical even though LLMs can evaluate a target model, which is fair. According to Table 3, foundation models and humans do not look consistent, and evaluation solely by the automated protocol seems insufficient. The paper seems faithful to this evaluation.

**Weaknesses:**

I think the dataset is useful for the application level, while it’s not clear from the technical level what aspects of a model it tries to evaluate. The proposed evaluation task seems to require (1) identification of the event, person, etc. depicted in the image, (2) associating the identified entities with the corresponding historical event (so that it can give a contextualized description), and (3) describing the image in a fair and objective way. I think (1) involves the perceptual capability of a model, while (2) and perhaps (3) involves the knowledge the model has. (3) may also involve the criterion of goodness of generated description used in the training. The proposed protocol evaluates a model without being aware of these different aspects (the paper partly mentions this point in Section 5.1), which makes the interpretation of the result extremely hard. I understand that as the foundation model users rarely have knowledge about how the model is trained, it’s not straightforward to isolate these different aspects. However, without some ways to interpret the results (as described in Section 5.1 as a future application of the dataset), insights that the dataset will provide may be limited.

The paper is often hard to read. I don’t see what the dataset offers (does it contain only images or some example descriptions of events?) in the main paper.

**Questions:**

I would like to see some discussion on the first point in the weaknesses.

---

> ### Author Response · Authors · 2024-11-15
> **Response to Reviewer zRQb**
>
> We thank the reviewer for thoughtful comments that scrutinise how  historical contextualisation –  a higher-level capability our benchmark claims to measure – may consist of multiple lower-level fundamental capabilities, including but not limited to entity recognition, scene understanding, and world knowledge.
>
> How different capabilities may be expressed when an AI system performs a complex task, such as historical contextualisation, is an important research question. We note that the  reviewer proposed unpacking into low-level tasks is not the only possible approach.
> In order to output a contextualised description of a historical image, such as a photograph from World War 2, a given model may be invoking any number of latent capabilities. Which capabilities are expressed and how they are encoded is likely to vary across model specifications, such as its technical architecture, training procedure and multimodal corpora it is trained on.
>
> Measuring specific lower-level capabilities has been the focus of many previous work (for example, MS-Celeb-1M for entity recognition, OKVQA for scene understanding, InfoSeek for world knowledge, Dollarstreet for fairness in object recognition, TallyVQA for object counting, DocVQA for OCR in context). While there is established coverage for low-level vision tasks, we identified a gap in the available evaluations that inspired us to create Century: there was no measurement for how well multi-modal models could contextualise and reason about images grounded in real-life events in open-ended text generation tasks.
>
> While our focus has been on measuring the higher-level historical contextualisation capability, we hope that releasing Century along with an evaluation protocol will enable additional studies of the interplay between low-level and high-level capabilities as pointed out by the reviewer.
>
> In this work, we propose a first decomposition which partially overlaps with the proposed (1) identification (2) association (3) fair / objective description breakdown.
>
> We target Capability (1),  the identification of the event or person depicted in the image, in our evaluation method as the “correct identification” question (Table 3). We ask auto-raters and human raters to identify if the target model correctly names the entity depicted in the image, providing a measure of the “perceptual capability” of the model.
>
> For (2), the reviewer makes an interesting point about the capabilities that may be necessary to associate figures and events depicted to their historical context. Qualitatively, we find that different images are more difficult to contextualise given the image alone (e.g. identifying the context behind a photograph of a famous political figure is more straight-forward than associating an image of a crowd with a specific historical event).  We discuss this point in the “Recommendation for Developers” section, but have pulled a longer discussion into results section to provide guidance on interpreting results in light of the differences in contextualisation difficulty (lines 425 - 473).
>
> For (3), or the “goodness” of the generation, this is covered by several of the evaluation criteria, including “factuality,” which evaluates if inaccuracies or factual errors are present in model output, and “appropriate summary,” which evaluates how much relevant detail on the event depicted is present in the model output.
>
> These partial overlaps in our decomposition approach and the approach recommended by the reviewer may indicate that the two approaches are complementary.
>
> To the reviewers’ concern on the paper not fully describing dataset contents, we have added  content in Section 3 to clearly state what fields are included in the datasets we are open-sourcing (lines 320-323). Alongside the publication, we will link the Github page, which will also describe the different fields contained in the dataset.
>
> We thank the reviewer for their careful engagement with the paper and their suggestions to strengthen the manuscript, which we have included in the updated version. If we have addressed the reviewer’s most pressing concerns, we kindly ask the reviewer to consider adjusting their score to reflect this. Otherwise, we look forward to proceeding with the discussion and incorporating any further feedback to our manuscript.

---

> > ### Comment · Reviewer_zRQb · 2024-11-26
> > **Thanks for the response**
> >
> > I appreciate the authors for detailed responses. I think the paper is valuable for the application, and I think the response makes sense. I'll update the score.

---

### Official Review · Reviewer_Ciz2 · 2024-11-03

**Soundness:** 4
**Presentation:** 3
**Contribution:** 4
**Rating:** 8
**Confidence:** 4

**Summary:**

This paper is about “Century” a dataset and framework designed to evaluate the ability of multi-modal models to contextualize sensitive historical images accurately, thoroughly, and objectively. To build the dataset, images were sourced with knowledge graphs, language models, and they were processed according to museum practices, considering especially recent historical events, figures, and locations, with images that may hold socio-cultural significance. The idea is to address the representation of historical nuance in generative AI and proposes an evaluation protocol for assessing models on tasks requiring socio-cultural and historical contextualization. After the construction of the Century dataset, it is tested with recent private foundation models. The paper reports that these models have difficulties addressing the complexity of historical contextualization.

**Strengths:**

- The proposed interdisciplinary dataset focus on historical contextualization and represent a valuable contribution to the field, addressing a crucial gap in existing evaluation methodologies.

- The work should be well reproducible with the released dataset, search terms, and evaluation details. Every part of the work is well detailed and released. Authors have put significant effort into this.

- The paper is well written, well structured and all parts, also detailed in the appendix, are well informative.

**Weaknesses:**

- the use of Wikipedia raises concerns about biases inherent of the platform. Wikipedia’s coverage of historical events is not uniform across regions or cultures, potentially leading to an overrepresentation of certain perspectives. Anyway, the limitation is acknowledged and is anyway a first step into the right direction.

- the definition of "sensitive" is based on interpretations from museums and archives, which seems a good starting point. However, I wonder about whose perspectives are considered "sensitive" and who gets to define them. Maybe some input from the communities whose histories are represented in the images should be considered, but I understand the difficulty of doing that.

**Questions:**

- Since the release of new LLMs are very frequent, I wonder what could be done to further automatise the evaluation on the dataset.

- I believe the dataset could potentially be misused to train models that generate biased or harmful content related to sensitive historical events. What do you think about this aspect?

- Could the limited representation of certain communities in the dataset be harmful for training of future models based on this dataset? I'm not sure about its inclusiveness and how to not perpetuate existing biases.

**Details Of Ethics Concerns:**

In my opinion:
- The dataset could potentially be misused to train models that generate biased or harmful content related to sensitive historical events.
- The limited representation of certain communities in the dataset could be harmful for training of future models based on this dataset, I'm not sure about inclusiveness and how to not perpetuate existing biases.

---

> ### Author Response · Authors · 2024-11-25
> **Response to Reviewer Ciz2**
>
> We thank the reviewer for their thoughtful review and appreciation of our contribution. We are pleased the reviewer recognizes the value of Century in addressing a key evaluation gap.
>
> Regarding the reviewer's concern about potential biases, we acknowledge that platforms like Wikipedia, from which we draw data, may introduce bias. However, our analysis (Table 8 in Appendix K) shows that Century includes images from all UN world subregions, with each represented by at least 5.1% of the total images. While there is an over-representation of Western Europe and North America, likely inherited from Wikipedia, Century still demonstrates a global reach.
>
> Initially, we intended to pursue direct collaboration with cultural heritage institutions and archives to increase representation for certain groups, but later found this was not feasible for this project. However, we believe Century provides a strong foundation for future partnerships and research with these institutions.
>
> We agree that incorporating participatory methods to assess image sensitivity is crucial, and we have highlighted this direction in our discussion under "Lack of targeted inclusion of affected communities." By releasing Century with geographical labels for each image, we hope to facilitate future research that includes participatory perspectives, including ethnographic studies focused on under-represented populations.
>
> In response to the reviewer's questions:
>
> * Automating Century evals: We demonstrate the performance of off-the-shelf foundation models as raters for the historical contextualization task. Table 3 (pg 3) demonstrates the directional alignment between the human ratings and autorater decisions. However, we believe more work is necessary to increase the accuracy and calibration of automated evaluation, and recommend retaining human evaluation for final model comparisons given the sensitivity and subjectivity of the task.
>
> * Potential misuse of the dataset: We believe the risk of misuse is low. We do not release any raw model outputs or human/auto-rater signals that could be used to train models to generate harmful outputs. Section 3 clarifies the specific fields included in the open-sourced datasets (lines 320-323).
>
> * Exacerbating biases through training: Century is not intended as a training dataset, so we do not anticipate it contributing to bias amplification in that way. However, we recognize the importance of improving dataset representativeness to enable effective evaluation of model performance across diverse communities.
>
> We thank the reviewer again for their thorough review and appreciate their view that this work offers a compelling contribution to the ICLR community.

---

### Meta-Review · Area_Chair_Yxbz · 2024-12-11

**Metareview:**

The paper proposes a dataset of ~1500 contextualized historical images, in order to test capabilities and modern methods in inferring this contextualization. The authors show existing methods do not handle this historical data well. The authors make an effort to cope with biases; some concerns about ethics and misuse, as well as choice of methods tested, are raised and addressed to a reasonable but not complete degree. The main contribution of the paper, to shed light on how historical or politically charged imagery may be described by current models, is significant despite these concerns.

**Additional Comments On Reviewer Discussion:**

Several reviewers responded that their concerns were addressed

---

### Decision · Program_Chairs · 2025-01-22

Accept (Spotlight)